# Nutrition as Prevention of Diet-Related Diseases—A Cross-Sectional Study among Children and Young Adults with Down Syndrome

**DOI:** 10.3390/children10010036

**Published:** 2022-12-24

**Authors:** Agnieszka Białek-Dratwa, Sebastian Żur, Katarzyna Wilemska-Kucharzewska, Elżbieta Szczepańska, Oskar Kowalski

**Affiliations:** 1Department of Human Nutrition, Department of Dietetics, Faculty of Health Sciences in Bytom, Medical University of Silesia in Katowice, Jordana 19, 41-808 Zabrze, Poland; 2Faculty of Health Sciences in Bytom, Medical University of Silesia in Katowice, 41-808 Zabrze, Poland; 3Internal Diseases and Diabetology Ward, Neurology Subdivision Specialist Hospital, no. 1 ul. Żeromskiego 7, 41-900 Bytom, Poland

**Keywords:** children, Down’s syndrome, diet, nutrition, dietary errors

## Abstract

Background: The aim of this study was to assess the diet of children with Down syndrome (DS) and to identify potential dietary mistakes made by their parents. Materials and methods: The study was conducted among 195 parents of people with DS between November 2020 and March 2021. Data for the study were collected anonymously using the CAWI method. Results: 122 (62.6%) people with DS did not eliminate any nutrient from their diet. By contrast, in the study group, the following numbers of people reported the following dietary restrictions: 51 (26.2%) gluten, 56 (28.7%) lactose, 17 (8.7%) casein, 26 (13.3%) sucrose, 2 (1.0%) histamine, 2 (1.0%) lectins, and 1 (0.5%) dairy. The most frequent response for vegetable and fruit consumption was once a day, with 83 (42.6%) and 87 (44.6%) parents indicating this. The most frequent response for dairy product consumption was every day, with 72 (36.9%) parents indicating this, while 36 (20%) parents stated that their children do not eat dairy products at all. In the study group, the most frequent response for meat consumption was several times a week, this was indicated by 107 (54.9%) parents, while 1 (0.5%) of them said that their children do not eat meat products at all. The most frequent response for fish consumption was 1–2 times a week, this answer was indicated by 101 (51.8%) parents, while 13 (6.7%) said that their children do not eat these products at all. Conclusions: A majority of the subjects with DS are usually fed in a normal way, but nutritional mistakes are made by the parents. Special attention should be paid to prolonging the period of natural feeding.

## 1. Introduction

Down syndrome (DS) is a set of birth defects caused by the presence of extra genetic material in the form of chromosome number 21. It is the most common chromosomal disorder among live-born infants. Its incidence ranges from 1 in 700–1500 live births [1]. The syndrome was first described in 1866 by a physician of British origin, John Langdon Down, after whom the syndrome is named [2]. Trisomy of chromosome 21 results in several defects, diseases, and dysfunctions due to the presence of this genetic mutation. DS is associated with an increased risk of medical problems, including gastrointestinal, cardiac, and pulmonary anomalies, as well as developmental delay and endocrine disorders [3]. In many cases, an appropriate dietary therapy is necessary, consequently avoiding the occurrence of additional conditions, such as becoming overweight or obese, which people with DS have a higher risk of than healthy individuals [4]. As shown in some clinical studies, children and adolescents with DS are often characterised by mild or moderate obesity [5,6,7,8], have higher levels of total fat mass, and fat mass is distributed in specific parts of the body compared with peers without DS [9,10].

Among the endocrine disorders that accompany DS, thyroid dysfunction is the most common. It is estimated to occur in 4–8% of children with DS [11]. The spectrum of thyroid dysfunction includes congenital hypothyroidism, subclinical hypothyroidism, acquired hypothyroidism (autoimmune/non-autoimmune), and hyperthyroidism [12]. Correct functioning of the thyroid gland is very important in the development of any child, especially a child with DS, as it ensures normal physical and mental development [13]. Thyroid diseases also occur in about 20–50% of adults with DS. This group of individuals is much more likely to have abnormalities of the thyroid gland compared with healthy individuals. Other studies indicate that hypothyroidism occurs in up to 46% of children with DS, depending on the age of the patient group and the study population [3]. Uncompensated hypothyroidism lowers the metabolic rate by up to 30–40%, which can cause the patient’s weight to increase and lead to obesity. Nutritional treatment plays a very important role in the treatment of hypothyroidism. The diet should be as balanced and varied as possible. It should be arranged according to the patient’s preferences, removing the mistakes he/she has made [14]. The caloric intake should be adjusted to the patient’s body weight; in the case of overweight or obese patients, a reduced-calorie diet should be used. Care should be taken to ensure an adequate supply of iodine, as it is necessary for the production of thyroid hormones and its deficiency may exacerbate iron, selenium, and retinol deficiencies [14,15,16]. Brassica vegetables hinder iodine absorption from food, so they should be limited [15,17]. An adequate supply of selenium is also necessary, as it is a component of the enzyme deiodinase and takes part in the conversion of thyroxine into triiodothyronine [14,18]. With limited selenium intake, triiodothyronine production is reduced and oxidative damage to the thyroid gland occurs; iodine absorption is also reduced [14]. Thyroid iodinating peroxidase, of which iron is a component, is responsible for activating the change of thyroglobulin to T4 and T3 [14,19]. Low iron levels pose a risk of decreasing triiodothyronine levels and increasing thyrotropic hormones [14].

In a study by Goday-Arno et al., the prevalence of hyperthyroidism in the DS population was 6.5 cases/1000 individuals [20]. Most studies on the prevalence of hyperthyroidism in the DS population indicate a low prevalence, generally below 3%, although the prevalence is higher than in the general population [21,22]. In this disease entity, one should remember to increase the caloric content of the diet by 15–20%, and in the case of advanced hyperthyroidism, even by 50–80%. The proportion of protein in the diet should also be increased (1.2–1.7 g/kg per day), as well as vitamin B_1_, antioxidant vitamins, and calcium. The supply of fats, including saturated fatty acids, trans-fatty acids, caffeine, and highly processed products low in nutrients should be limited [14].

Autoimmune disorders are more common in patients with DS compared with the general population [1]. Among the reported autoimmune disorders in this group of patients, the most common are celiac disease, with a prevalence of 5–10%; type 1 diabetes, which is thought to be three times more common in DS patients [3]; alopecia, with a prevalence of 11.4% [23]; and autoimmune thyroid disease [3,24].

Thyroid autoantibodies are detected in 13–34% of patients with DS [3]. Antibodies to thyroid peroxidase (TPO) have been detected in up to 31% of patients with DS [25]. The presence of TPO antibodies strongly correlates with the evolution from euthyroidism and subclinical hypothyroidism to overt hypothyroidism [24]. Autoimmune hypothyroidism or Hashimoto’s thyroiditis is more common than hyperthyroidism or Graves’ disease [1,23]. Autoimmune hypothyroidism in DS is equally common in both sexes, in contrast to the predominance of women with this condition in the population without DS [1,3]. Treatment consists of levothyroxine pharmacotherapy at a dose selected according to the degree of hypothyrosis, the coexistence of other conditions that alter the absorption of the drug in the gastrointestinal tract as a result of the course of treatment, and one that maintains normal TSH levels in the blood [3,26].

In Hashimoto’s thyroiditis, diet therapy is important in addition to pharmacology, but there are currently no established standards for nutritional management [3,14,26]. Diet therapy for Hashimoto’s disease is used to slow down the inflammatory process in the thyroid gland and its destruction. It is advisable to increase the supply of omega-3 fatty acids and antioxidant vitamins which show anti-inflammatory effects, such as vit. C, E, A. The protein intake in this group of patients should reach even 25% of the total energy requirement. An adequate supply of vitamin D [27,28] and B12 should also be taken care of, due to the frequent co-occurrence with autoimmune thyroiditis of malignant anaemia [14].

Celiac disease affects approximately 7–20% of individuals with DS [29]. In a meta-analysis, Du et al., analysing over 30 studies, found that in Europe and America, individuals with DS are at very high risk of celiac disease, with more than one in twenty patients (children) with DS having celiac disease [30]. In a Polish study, patients with DS were a high-risk group for celiac disease, with a prevalence estimated to be at least 5.4% [31]. Celiac disease is an autoimmune systemic disease consisting of an abnormal reaction of the organism (allergy) to gluten, found in certain grains, e.g., wheat, barley, and rye. Common symptoms of visceral disease include gastrointestinal disorders, such as bloating, abdominal pain, malabsorption, and weight loss. In some cases, chronic fatigue, microcytic anaemia, epilepsy, depression, susceptibility to miscarriage, osteoporosis, osteopenia, vitamin deficiencies, enamel defects, delayed puberty, and autoimmune diseases occur [29,31]. Dietary therapy for coeliac disease consists of the elimination of each food item containing gluten. Depending on the degree of damage to the intestinal villi, the appropriate level of digestibility of the diet should be adjusted [29].

Down syndrome is characterised by typical dysmorphic features, hypotonia, delayed physical and psychomotor development, mental retardation, and the coexistence of gastrointestinal defects, increased risk of leukaemia, hypothyroidism, and congenital heart defects, among others, which occur in approximately 40–63% of children with DS [11,32,33]. Among congenital heart defects occurring in children with DS, non-sinusoidal defects with increased pulmonary flow due to left–right leakage predominate. Among other things, they predispose to more frequent lower respiratory tract infections and lead to symptoms of heart failure, to a degree that depends on the type of defect and the severity of the haemodynamic disturbance. Non-sinusoidal heart defects, which cause a significant increase in pulmonary flow, consequently lead to the development of pulmonary hypertension and Eisenmenger syndrome. A non-sinus heart defect then becomes a sinus defect and closes the possibility of reparative cardiac surgery [34].

The spectrum of heart defects in children with DS varies somewhat according to genetic, geographical, and sociodemographic factors. In Western European countries and the U.S.A., 43% of children with DS are most commonly found to have atrioventricular subendocardial defects and atrioventricular septal defects (AVSD), 32% have a ventricular septal defect (VSD), 10% have the atrial septal defect of the secondary orifice type (ASD II), 6% have Fallot syndrome, and 4% have isolated patent ductus arteriosus (PDA) [34]. Approximately 30% of children with DS have several co-occurring heart defects. The most common heart defect in children with DS, with the highest risk of developing heart failure and pulmonary hypertension, is total AVSD. A congenital heart defect is also recognised as a manifestation of DS and is associated with a higher risk of cardiogenic embolism [35].

Causes of the development of overweight and obesity in DS include hypotonia (decreased muscle tone), susceptibility to systemic inflammation, metabolic disease, and/or metabolic slowing [36]. Typically, people affected by DS eat fewer healthy foods, show physical limitations, depression, and lack social and financial support. In addition, medication contributes to weight gain [37].

Therefore, proper diet and physical activity are very important for Down syndrome. The World Health Organisation [38] recommends that children should carry out at least 60 min (of MVPA) per day. Unfortunately, meeting the recommended level of physical activity can be a challenge for children with disabilities. Furthermore, meeting the recommended daily MVPA becomes even more challenging for children with Down syndrome, as they are at risk of physical inactivity and obesity [39]. Therefore, strategies are needed to improve the quantity and quality of their physical activity. Studies have been conducted in which facilitators of physical activity for children with DS were analysed. Three commonly cited facilitators of physical activity that encourage children with DS to participate more frequently in daily activities are the positive role of the family, social interaction with peers, and available programmes that provide adaptations for children with DS [40,41]. Parents play a central role in encouraging and supporting physical activity for their children with DS [42].

Proper nutrition for people with DS is a very important but equally difficult task. Difficulties may result from many factors, e.g., coexistence of the above-mentioned diseases and the occurrence of mental disabilities of different severity [4]. The diet should take into account all diseases of the patient and each patient should be approached individually. The proposed dishes should be by the patient’s preferences, as it is mainly the patient who decides whether to eat a given meal.

The study aimed to assess the diet of children with DS and to identify potential dietary errors made by parents.

## 2. Materials and Methods

### 2.1. Study Group

The survey was conducted among 211 Polish parents of people with DS between November 2020 and March 2021. Data for the study were collected anonymously using the Computer-Assisted Web Interview (CAWI) method; an online questionnaire was distributed in forums and support groups for parents of people with DS. Respondents were informed about the purpose of the study and how the results would be used, after which they agreed to participate. Conducting the survey did not require the authors to obtain approval from a bioethics committee in light of the Act on Physician and Dentist Professions of 5 December 1996, which includes a definition of medical experimentation.

For the final analysis, considering the inclusion and exclusion criterion, information collected from 195 individuals was considered.

### 2.2. Inclusion and Exclusion Criteria

The inclusion criteria for the study were legal guardianship of a person with DS, consent to participate in the survey, and filling out the questionnaire correctly and completely.

Criteria for exclusion from the study were enteral or parenteral feeding of the child, situations where the parent is not responsible for what the child is fed (child is in care), and incorrect or incomplete completion of the questionnaire.

### 2.3. Research Tool

The research tool used in the study was an original survey questionnaire. The questionnaire contained 30 questions, both closed and open. The questionnaire was validated.

The first part of the questions focused on the metric with socio-demographics of the parents/carers (gender, legal custody status, age, education) and their children (gender, child’s age in years and months, child’s height/length, child’s weight, feeding method in the first six months of life—natural feeding, mixed feeding, milk formula feeding). Associated diseases, such as type 1 diabetes mellitus (insulin-dependent), type 2 diabetes mellitus (insulin-independent), hypothyroidism, hyperthyroidism, Hashimoto’s, coeliac disease, duodenal atresia, anal atresia, Hirschsprung’s disease, lactose intolerance, oropharyngeal dysphagia (upper, transoesophageal), and oesophageal dysphagia (lower), were also included in the study.

In the second part of the study, dietary habits were assessed. The dietary assessment considered the frequency of consumption of individual products with specific foods, such as vegetables, fruits, sweets, nuts, meat, type of meat consumed most often, fish, dairy, type of dairy consumed, and eggs. The assessment of the frequency of individual products with specific food was compared with recommendations for the frequency of consumption of these products per day. Per week—vegetables, fruits, and dairy should be consumed several times per day, fish 1–2 times per week, meat and eggs several times per week, while with sweets the less frequent, the better. The following scaling was used in the study—dairy, fish, eggs (daily, 5–6 times a week, 3–4 times a week, 1–2 times a week, once every 2 weeks, once every 3 weeks, once a month, less often, does not consume) meat (several times a day, once a day, 5–6 times a week, 3–4 times a week, 1–2 times a week, does not consume), fruits and vegetables (several times a day, once a day, 4–6 times a week, 1–3 times a week, less often, does not consume) and sweets (several times a day, once a day, 4–6 times a week, 2–3 times a week, once a week, once every 2 weeks, once every 3 weeks, once a month, less often, does not consume).

The dietary assessment considered eliminating foods containing gluten, lactose, sucrose, casein, histamine, and lectins. The number of daily meals consumed from 1 to 7 or more was also assessed.

A pilot study was carried out to validate the questionnaire and to check the relevance and acceptability of the questions contained in it. The survey was conducted on a sample of individuals representing the population of mothers. The pilot study was conducted among 20 mothers; after one month, the questionnaire was administered again to the same group (pilot study 2).

The reproducibility of the responses was examined by comparing the responses from the pilot study and pilot study 2. Pilot study 2 took place one month after the study to avoid the freshness effect. To assess the reproducibility of the results obtained with the questionnaire used, the ϰ (Kappa) parameter was calculated for each question of the questionnaire (results obtained in the pilot and pilot study 2)—for 72.6% of the questions, very good (ϰ ≥ 0.80) concordance of answers was obtained, while for 18.6% of the questions, good (0.79 ≥ ϰ ≥ 0.60) concordance of methods was obtained. For 8.8% of the questions in the questionnaire analysed, the concordance between the results obtained at baseline and retest was moderate (ϰ < 0, 59).

The questionnaire was divided into information concerning the examined parent (mother/father/legal guardian, the age of parent/guardian, education of parent/guardian); metric data of the concerned child (sex, age, weight, height); information concerning the method of feeding during infancy (natural feeding/formula feeding, length of breastfeeding, weight gain during infancy); coexisting diseases in the examined child; supplementation (dietary eliminations in the daily diet of the child, frequency of consumption of specific food groups and beverages, length of breastfeeding, infant weight gain).

### 2.4. Analysis of Results

Age-appropriate BMI percentile grids were used to estimate normal body mass in children (percentile grid and 3 SD BMI of girls aged 0–3 years; WHO standard; percentile grid and 3 SD BMI of boys aged 0–3 years; WHO standard; centile grid and SDS BMI and the limits of underweight, overweight and obesity for girls aged 3–18 years; OLA and OLAF study; centile grid and SDS BMI and the limits of underweight, overweight and obesity for boys aged 3–18 years; OLA and OLAF study) [43,44,45], and for adults, BMI body mass index [46]. According to the recommendations of the Centre for Child Health, underweight among children is considered when the body mass index is between 0 and 10 percentile BMI, normal body weight is assumed between 10–85 percentile, overweight 85–95, obesity 95–100 [44,45]. According to the current WHO recommendations, underweight among adults was found at BMI values below 18.5 kg/m2, normal weight, overweight, and obese at values of 18.5–25 kg/m2, 25–30 kg/m2, and above 30 kg/m2, respectively [46].

Microsoft Office Word, Microsoft Office Excel, and Statistica 13.0 were used to analyse the collected data.

## 3. Results

### 3.1. Characteristics of the Study Group of Parents/Carers and People with Down Syndrome

Of those participating in the study, the majority were female (*n* = 188, 96.4%). In terms of age, the largest group were parents aged 36–40 years (*n* = 56, 28.7%), followed by 41–45 years (*n* = 42, 21.5%) and 31–35 years (*n* = 37, 19.0%) and those with a master’s degree (*n* = 77, 39.5%), followed by those with secondary education (*n* = 61, 31.3%) and a bachelor’s degree/engineering degree (*n* = 40, 20.5%) (Table 1).

Analysis of the obtained results showed that among the subjects with DS, there were *n* = 98 (50.2%) females and *n* = 97 (49.8%) males. The mean age of the study group was 6.6 years ± 5.8 years (min 0.5 years–max 30 years; median 4.5 years). Due to the large age range, the cohort was divided into age groups: nursery-aged children 0.5–2.5 years (*n* = 63, 32.3%), preschool-aged children 3–6.5 years (*n* = 65, 33.3%), school-aged children 7–10.5 years (*n* = 24, 12.3%), adolescents 11–17.5 years (*n* = 35, 17.9%), and a group of adults over 18 years (*n* = 8, 4.1%) (Table 2).

Of the total group, 41 (21.0%) were underweight, 113 (57.9%) were normal weight, 30 (15.3%) were overweight, and 11 (5.6%) obese. In 70 (35.9%), no diseases other than Down syndrome were found. However, among the associated diseases mentioned, hypothyroidism (*n* = 102; 52.3%) and lactose intolerance (*n* = 19; 9.7%) were most frequently indicated.

The analysis of the results further showed that 131 (67.2%) children were breastfed. In 54 (27.7%), there was no possibility of breastfeeding for medical reasons, and 10 (5.1%) were not breastfed, although such a possibility existed. The most frequently indicated period of breast milk feeding was 7–12 months with 32 (16.4%) children, followed by 1–3 months with 26 (13.3%), and 4–6 months and 13–18 months with 21 (10.7%) children each. Moreover, only 11(5.6%) are fed according to the dietician’s recommendations (Table 2). The elimination of specific components from the diet was also included in the study. A total of 122 (62.6%) subjects with DS did not eliminate any nutrient from their diet, 51 eliminated (26.2%) gluten, 56 (28.7%) lactose, 17 (8.7%) casein, 26 (13.3%) sucrose, 2 (1.0%) histamine, 2 (1.0%) lectins, and 1 (0.5%) dairy.

### 3.2. Intake Frequency of Selected Food Groups among Persons with Down Syndrome

Figure 1 and Figure 2 show the frequency of vegetables and fruits consumption in the study group of individuals with DS. As the analysis showed, vegetables and fruits are consumed most frequently with a frequency of once a day; such answers were indicated by 83 (42.6%) and 87 (44.6%) parents, respectively, and 4 (2.1%) and 3 (1.5%) of them indicated that their children do not eat these products, respectively.

Dairy products are food products containing milk from farm animals and products derived from milk (cheese, yoghurt, kefir, etc.). The analysis of the results showed that dairy products are consumed most frequently daily; this answer was indicated by 72 (36.9%) parents, while 36 (20%) of them stated that dairy products are not eaten by their children at all (Figure 3). An important issue is the type of dairy products consumed. The best choice is natural dairy products without added sugar, colourings, etc. The results showed that for 33 (16.9%) of the respondents, milk is the primary type of dairy consumed, followed by fermented milk products 64 (32.8%), homogenised cheese 30 (16.3%), cottage cheese 11 (5.6%), and yellow cheese 18 (9.2%).

Eggs are a rich source of protein and contain many vitamins. The analysis showed that eggs are most often eaten 1–2 times a week; this answer was indicated by 76 (39%) parents, while 22 (11.3%) answered that their children do not eat these products at all (Figure 4).

Analysis of the results showed that meat is consumed most frequently several times a week; such an answer was indicated by 107 (54.9%) parents, while 1 (0.5%) of them said that meat products are not eaten by children at all (Figure 5). The most commonly consumed meat in the study group was chicken (*n* = 155, 79.5%), turkey (*n* = 109, 55.9%), and pork (*n* = 101, 51.8%) (more than one answer possible).

Fish is a valuable source of, among others, protein, and omega-3 fatty acids (e.g., EPA, DHA). The analysis showed that fish is most often eaten 1–2 times a week, which was the answer given by 101 (51.8%) parents, while 13 (6.7%) responded that their children do not eat fish at all (Figure 6).

The analysis showed that 24 (12.3%) people do not give their children nuts for health reasons (allergy), and 141 (72.3%) do not give nuts without justification. The remaining persons regularly eat nuts (Figure 7).

Most parents of persons with DS indicated still water as the most commonly consumed drink by their children—103 (52.8%) persons, followed by 34 (17.4%) fruits juices, 16 (8.2%) sweetened tea, and 12 (6.2%) vegetable juice. Sweetened carbonated drinks were very rarely served in the study group.

The analysis also showed that 64 (32.8%) of the respondents do not give sweets to their children at all, while out of 131 parents who give sweets to their children several times a day, they give them 2 (1.0%), once a day 18 (9.2%), 14 (7.2%) 4–6 times a week. A total of 53 (27.2%) children consume sweets once a week or less.

The analysis of the results showed that parents usually give their children 5 or 4 meals a day, as declared by 103 (52.8%) and 51 (26.2%) parents, respectively (Figure 8).

## 4. Discussion

Nutrition for people with Down’s syndrome is rarely discussed in the literature. However, it is an essential issue because, if carried out correctly, it improves normal development in most cases and facilitates the treatment of associated diseases.

An essential part of the development of every child, and especially of a child with Down syndrome, is the length of natural breastfeeding, preferably for a minimum of 2 years. The composition of breast milk changes over time so that it is constantly adapted to the needs of the growing organism. Breast milk contains all components necessary for normal growth, including proteins, fatty acids, carbohydrates, vitamins, and minerals [47]. The first attachment to the breast should begin no later than 2 h after birth. Healthy babies should be put to the breast as often as possible, depending on their needs, and at night. A child fed on demand can regulate the intervals between feeds on its own [48]. As far as children with DS are concerned, however, for some time after birth they are characterised by low activity and may oversleep at feeding times. Therefore, they should be fed only on demand and woken up at mealtimes [47,48].

The World Health Organization (WHO) recommends exclusive breastfeeding until 6 months of age and then continuing with complementary foods until the child reaches 2 years of age or longer, all depending on the mother’s and child’s desire [49]. In our study, 43.51% of the children with DS studied (among all *n* = 131 naturally fed children) were breastfed for less than 6 months, and a total of 90.08% of the children studied were breastfed for less than 24 months. This is too short a duration of breastfeeding, which in the case of children with DS, may negatively affect their development [49].

In people with DS, food elimination should not be used when there are no absolute indications, i.e., malabsorption and digestion disorders and food intolerance or allergy [4]. In the case of most study subjects, it was noted that dietary elimination of nutrients is used without clear reasons. Analysis of the results showed that 19% of people (9.7) were diagnosed with lactose intolerance. Still, the percentage of people who exclude lactose from the diet is 28.7% (56), which means that only 33.93% of people do so justifiably. A diet low in lactose, in the absence of the occurrence of intolerance, is completely unjustified. Moreover, such a diet may consequently lead to the disappearance of the production of the enzyme lactase that causes lactose intolerance.

The main source of water is in the diet drinks. Some of them, apart from water, contain other unnecessary products, such as sugar, sweeteners, flavour enhancers, etc. Therefore, it is advisable to choose natural still water for drinking and occasionally natural 100% juices or sugar-free drinks with natural composition [50]. The main source of water in the diet of adolescents in the study by Marcinkowska U. et al., is mineral water—96% of the respondents declared its consumption [51]. In our study, most respondents (52.82%) with DS also choose natural still water as the main beverage. This is less than in the above study. This may be because most DS subjects studied are children who usually do not like pure water. Hence, parents, for their convenience, give their children juices or other drinks that are tasty without thinking about the future health consequences of their decision. Another good source of water can also be unsweetened tea (3.59%) or vegetable juice (1.54%), which additionally contains vitamins and minerals.

Sweets are products that children, as well as adults, very willingly eat. In the study by Kmiecik D. et al., 91% of Poles admitted to buying sweets, and most acknowledged that they are an essential part of their diet [52]. In the present study, 56.41% of the respondents admitted to buying sweets, but only 6.15% admitted eating sweets bought only in a shop. This difference may be due to this group of respondents’ more excellent knowledge of healthy eating and the prevention of metabolic diseases, to which people with DS are more vulnerable than healthy individuals.

Harton A.’s study argues that many mistakes are still made in children’s nutrition and emphasises there is a lower intake of vegetables compared to fruits, which are more tempting in their taste [53]. Insufficient intake of vegetables and fruits was noted in her study by Maria Gacek; she found a similarly higher intake of fruits than vegetables [54]. The results of our study indicate that about ¼ of the respondents consume vegetables and fruits too rarely. The reason for this may be insufficient knowledge of the parents about the benefits of proper nutrition or the children’s aversion to the taste of certain vegetables or fruits.

Nuts are considered a valuable source of protein, fat, and dietary fibre. They are rich in energy and nutrients. As shown by some studies, their regular consumption may reduce the risk of certain diseases, e.g., cardiovascular diseases and cancer. Nuts have also been shown to enhance cognitive development [55]. A low glycaemic index characterises them due to their high content of unsaturated fatty acids, protein, and relatively low carbohydrate content. In recent years, there has been an increased interest in nuts as an essential part of a healthy diet, given their positive effects on health [56,57]. Unfortunately, in our study, the consumption of nuts was proven below, with as many as 72.31% of the respondents not consuming them regularly without a valid reason. Nuts contain tocopherol, phenolic compounds, and selenium, which have strong antioxidant properties, and oxidative stress is associated with the development and ageing process of individuals with DS [58,59].

Proteins from products of animal origin, e.g., dairy products, eggs, poultry meat, and fish, are considered to be excellent and complete sources of protein [60]. According to the principles of healthy nutrition, each meal should contain an ingredient that is a source of animal protein. Meat additionally contains vitamin B12, the deficiency of which may lead, among others, to macrocytic anaemia. Our study showed that almost all respondents give meat to their children, but only 48.72% consume it with sufficient frequency.

Chicken, turkey, rabbit, lamb, and veal can be considered the most valuable types of meat. In our study, respondents indicated chicken (79.49%), turkey (55.90%), pork (51.79%), and rabbit (21.54%) as the most frequently chosen type of meat, which is a good choice. In the case of pork, however, it should be noted that it is advisable to choose lean cuts, such as pork loin.

In Poland, fish are usually eaten occasionally, and they are the best source of omega-3 fatty acids [61,62]. They are also rich in complete protein, vitamins, and minerals. They should be consumed at least 1–2 times a week. In our study, 54.87% of respondents follow these recommendations, 38.45% consume them less frequently, and 6.67% do not consume fish at all. Research conducted by Dymkowska-Malesa M. et al. [63] among adolescents has shown that more than half of the respondents consume fish according to current dietary recommendations. However, the overall fish consumption in Poland is relatively low, although fish are well-promoted in the media.

Enriching the daily diet with eggs is beneficial for health. They are very well balanced regarding nutrient contribution to energy value [64]. Although the American Heart Association recommends limiting dietary cholesterol intake, it does not suggest limiting egg consumption. Recent scientific studies confirm that their consumption may reduce the risk of certain diseases [65]. The relationship between egg consumption and mortality from coronary heart disease is significant [66]. Analysis of the results of our study showed that most of the subjects need to consume eggs more frequently, despite the benefits of their consumption. This may be due to needing more knowledge of the benefits of egg consumption.

Recommendations for the diet of people with Down syndrome are primarily concerned with considering the comorbidities that determine the nutritional basis. The prevalence of overweight and obesity in DS ranges from 23 to 70% in both males and females and is due to altered food intake, reduced physical activity, and a basal metabolism lower than in the general paediatric population [4,5]. The importance of diet in DS is also a severe problem of overweight and obesity among these individuals. Therefore, a comprehensive programme of nutrition and physical activity tailored to the individual’s specific needs must be addressed. Weight gain can be caused by hypothyroidism or a combination of inactivity and poor eating habits. In their daily diet, people with DS should eat a variety of vegetables, whole grains, low-fat dairy products, lean protein, and fruits. They should not over-snack, especially on products containing sugar and fats, and incredibly saturated and trans fatty acids. Sweetened beverages should be eliminated from the diet in favour of increased water consumption [4,7,8,59]. Considering the above, people with DS should eat regular meals, small in volume, and eat vegetables and fruits as often as possible, with more vegetables. These products are the primary source of vitamins (especially vitamin C, beta-carotene, and folate), minerals, fibre, and natural antioxidant—so-called antioxidants—which remove free oxygen radicals that are harmful to the body. Eating fruits and vegetables regularly reduces the risk of developing many diseases, including type 2 diabetes, obesity, high blood pressure, and ischaemic heart disease. It is best to eat them raw or minimally processed, as the products retain the highest nutritional value in this form. Vegetables and fruits should be eaten several times daily for meals and snacks. Eat whole grain cereals containing small amounts of protein and fat, B vitamins (essential for the nervous system, concentration and learning), and minerals (e.g., magnesium, zinc, iron), which support physical development and improve mood and learning ability. They are also a source of dietary fibre, which supports the digestive system, helps maintain average body weight, and helps prevent diseases caused by poor nutrition. Eat fish, pulses, and eggs. Choose lean meat. Limit the consumption of processed meat products, such as cold cuts and sausages. Avoid the consumption of sugar, sweets, and sugary drinks. Replace them with fruits and nuts. Limit the intake of animal fats. Replace them with vegetable oils [4,5,7,8,59]. In Appendix A, we have proposed the most important dietary recommendations for people with Down syndrome.

## 5. Conclusions

After analysing the results, the following conclusions were drawn:Most Down syndrome subjects are usually well fed, but parents make dietary mistakes. Particular attention should be paid to extending the period of natural feeding.In the study group of children and adolescents with Down syndrome, the frequency of consumption of dairy products was insufficient (only 36.9% consumed dairy products daily). Meat was consumed too frequently (more than 40% of the subjects consumed it daily or several times a day), while fish consumption was insufficient, as was the consumption of nuts. Only 31.8% of the respondents consumed enough vegetables each day, with better results for fruits consumption, with 37.4% consuming it several times a day, and 44.6% once a day. Considering all the analysed results, it is necessary to include nutrition education for parents/guardians in the therapeutic process so that they can properly and skilfully compose the daily diet of children/young people with DS.The abnormalities may be due to insufficient knowledge of parents about proper nutrition. Appropriate nutritional education of parents and/or persons responsible for nutrition is needed, which may contribute to improving their health and well-being.It seems essential to educate parents of children with Down syndrome so that they can feed their children appropriately, taking into account their needs for the prevention of overweight and obesity and those associated with other co-morbidities, e.g., cardiomyopathies, hypothyroidism, coeliac disease, allergies, and food intolerances.

## 6. Study Limitations

The results of our study should be interpreted with its limitations in mind. The parents provided all information, which may cause information bias. However, given the specificity of the study group and the timing of the study, another form of information collection was difficult.

The survey was conducted using the CAWI method, which was criticised for needing more insight into the data collection process. However, it is worth noting that this data collection method is widely accepted and convenient for collecting large amounts of information in groups that are often difficult to access. The survey was conducted nationwide (Poland) and cannot be referred to other populations worldwide.

An advantage of the study is the large group of 195 parents of people with DS, where most of the group consists of nursery, pre-school, and early school-age children. Very little research has been conducted on this topic. Hence, the discussion draws on research that is 30 years old.

The limitation of our study was the period of the second and third waves of SARS-CoV-19 in Poland and its partial lockdown. Due to the partial lockdown, many people, including the parents of our study subjects, worked partly remotely, which may have translated into the diet with dietary changes. However, with the introduction of the regulation, principals of special schools, special education centres and revalidation centres, as well as special schools in therapeutic entities and social welfare units, will be able to decide themselves on the mode of teaching—remote or stationary—during this period. For students who, due to a disability or home conditions, for example, could not study remotely at home, the head of the school was obliged to organise stationary or remote teaching at school (using computers and the necessary equipment located at school). Unfortunately, in our study, we should have considered staying at school or home, which may be a study limitation.

## Figures and Tables

**Figure 1 children-10-00036-f001:**
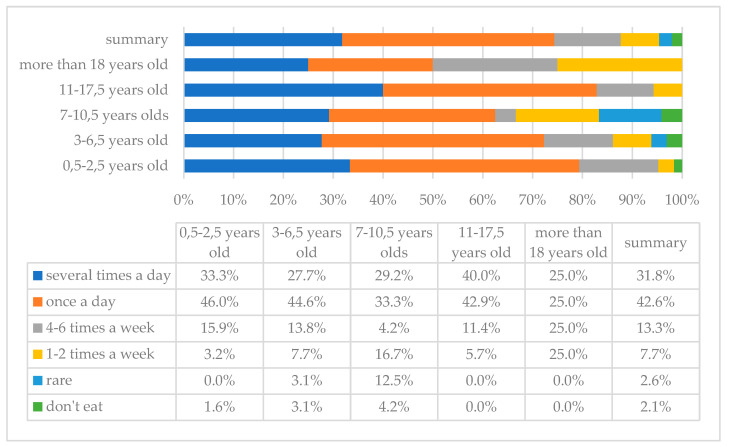
Frequency of vegetable intake in the study group of persons with DS with breakdown by age.

**Figure 2 children-10-00036-f002:**
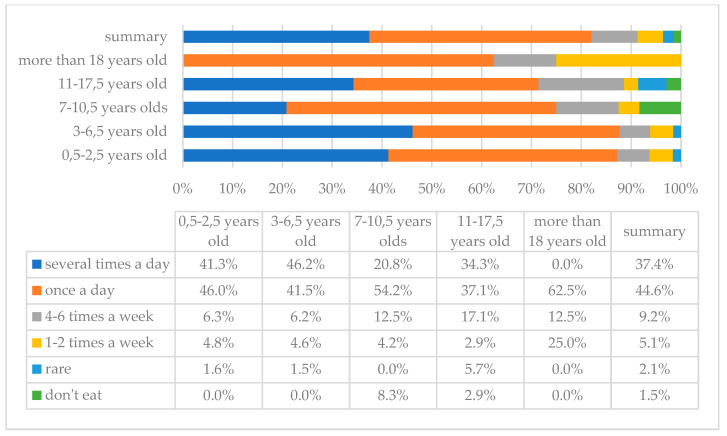
Frequency of fruits intake in the study group of individuals with DS by age.

**Figure 3 children-10-00036-f003:**
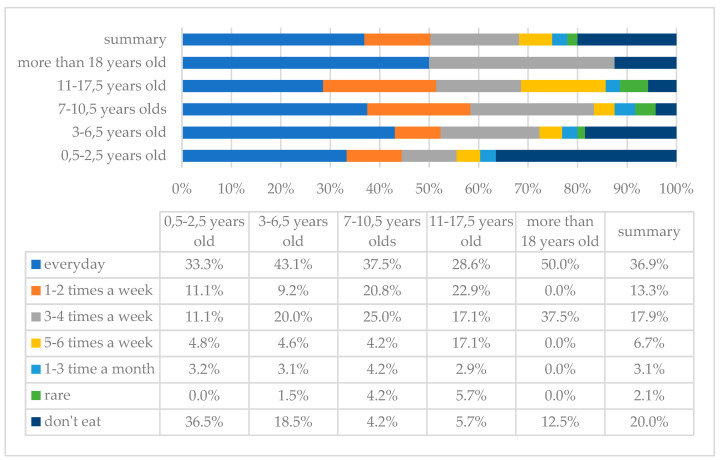
Frequency of dairy intake in the study group of individuals with DS by age.

**Figure 4 children-10-00036-f004:**
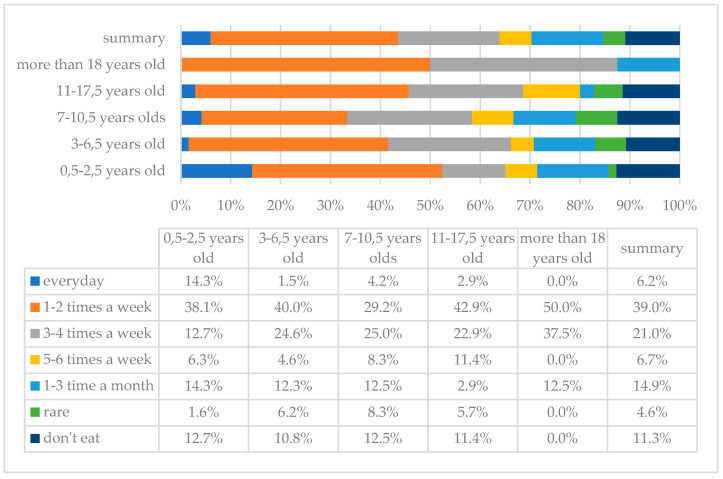
Frequency of egg consumption in the study group of individuals with DS by age.

**Figure 5 children-10-00036-f005:**
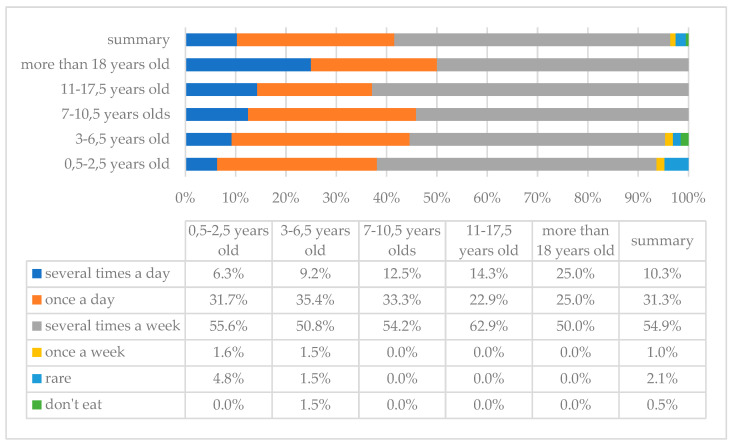
Frequency of meat consumption in the study group of individuals with DS by age.

**Figure 6 children-10-00036-f006:**
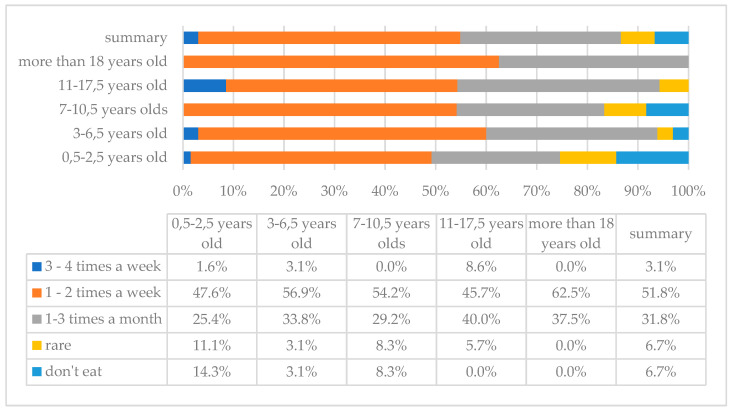
Frequency of fish consumption in the study group of individuals with DS by age.

**Figure 7 children-10-00036-f007:**
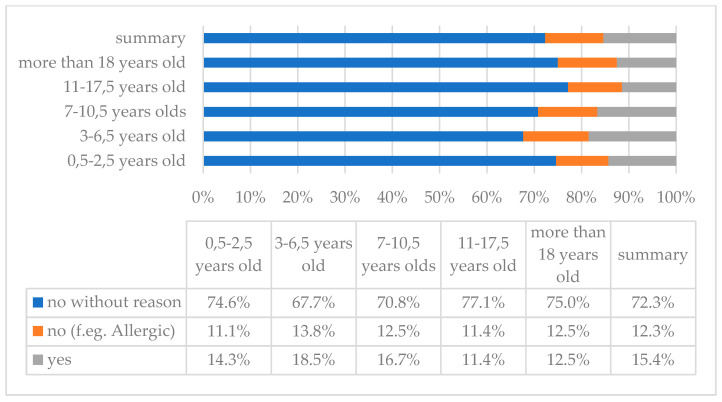
Nut consumption in the study group of people with DS by age.

**Figure 8 children-10-00036-f008:**
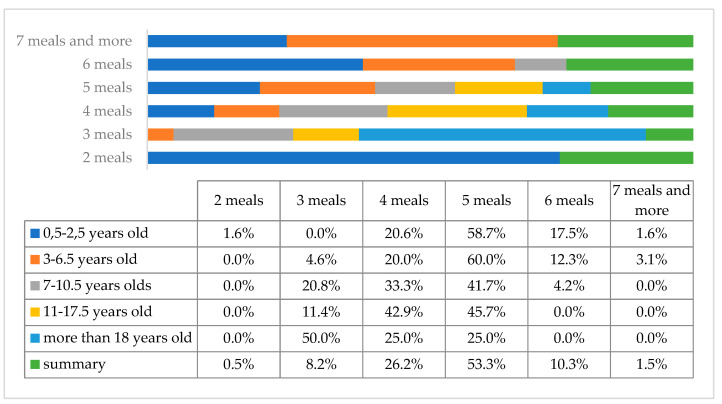
A number of meals consumed per day in the study group of individuals with DS by age.

**Table 1 children-10-00036-t001:** Characteristics of the studied group of parents.

	*n*	%
Gender of parent:		
woman	188	96.4%
man	7	3.6%
Parent’s age:		
18–30 years	30	15.4%
31–35 years	37	19.0%
36–40 years	56	28.7%
41–45 years	42	21.5%
46–50 years	20	10.3%
51 and over	10	5.1%
Parent’s education:		
basic	2	1.0%
medium	61	31.3%
professional	14	7.2%
higher bachelor’s engineering degree	40	20.5%
higher master’s degree	77	39.5%
higher doctoral studies	1	0.5%
total	195	100%

**Table 2 children-10-00036-t002:** Characteristics of the study group of participants with Down syndrome.

	*n*	%
Gender:		
female	98	50.2%
men	97	49.8%
Age by group		
0.5–2.5 years	63	32.3%
3–6.5 years	65	33.3%
7–10.5 years	24	12.3%
11–17.5 years	35	17.9%
18–30 years	8	4.1%
Bodyweight *		
underweight	41	21.0%
normal body weight	113	57.9%
overweight	30	15.3%
obesity	11	5.6%
Associated diseases		
no	70	35.9%
hypothyroidism	102	52.3%
lactose intolerance	19	9.7%
celiac disease	7	3.6%
Hashimoto’s thyroiditis	7	3.6%
oesophageal dysphagia (lower)	6	3.1%
oropharyngeal (upper) dysphagia	6	3.1%
hyperthyroidism	5	2.6%
Hirschsprung’s disease	5	2.6%
anal atresia	4	2.1%
duodenal atresia	2	1.0%
diabetes	3	1.5%
Breastfeeding		
yes	131	67.2%
it was not possible	54	27.7%
no, although it was a possibility	10	5.1%
Length of breastfeeding		
no	64	32.8%
less than 1 month	10	5.1%
1–3 months	26	13.3%
4–6 months	21	10.7%
7–12 months	32	16.4%
13–18 months	21	10.7%
19–24 months	8	4.1%
more than 2 years	6	3.1%
the child is currently still breastfed	7	3.6%
Following a diet arranged by a specialist:		
yes	11	5.6%
not	184	94.4%
Total	195	100%

* For children, BMI centile grids were used to estimate body weight normality for adults BMI.

## Data Availability

The data presented in this study are available on request from the corresponding author. The data are not publicly available due to restrictions that apply to the availability of these data.

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
