# Peer review of "Nutrition as Prevention of Diet-Related Diseases—A Cross-Sectional Study among Children and Young Adults with Down Syndrome"

_children, 2022, doi:10.3390/children10010036_

Round 1
Reviewer 1 Report
Review of the article: Nutrition as prevention of diet-related diseases - a cross-sectional study among children and young adults with Down Syndrome
The idea to study is good and necessary. However, this article needs some clarification and correction.
Major comments
· References are very few.
· Why were validated Food Frequency Questionnaires (FFQ) not used in your study?
Please explain your choice in the manuscript. Consider adding your survey questionnaire as supplementary material.
· Why has dietary intake (consumption of microelements and microelements in daily food ration) not been assessed? These results would provide more information on possible vitamin and mineral deficiencies discussed in the introduction.
· What were the criteria for the division into age groups of participants - were there any significant changes in the diet and food frequency of consumption during these periods?
· Consider adding statistical analysis where compare consumption frequency between age groups.
· In the discussion chapter, write if everyone with comorbidities followed a specific diet/treatment, e.g., celiac disease or diabetes.
· Conclusions should be redrafted to be based on the results obtained.
Minor comments
· Please correct editorial mistakes.
· Abbreviations should be specified in the first appearance in the text
· Lines: 23-24 - Please correct this sentence. - The sentence suggests that everyone with Down Syndrome should eliminate these nutrients.
· Table 2. Please correct the title of the table – “people” correct to “participants.”
· Why were the young adults included in the study (small number of participants)?
Author Response
Dear Reviewer.
Thank you very much for your valuable time. We have responded to all your suggestions below. In the text, we have highlighted any substitutions in yellow to make it easy to follow what we have added.
At the same time, we apologise that you had to wait so long for our response.
We did not use the full standardised FFQ in the survey, but used some of the questions from it. We relied on the Polish standardised questionnaire KomPAN, however, it is not defined to be used among children only for people aged 16 and over, hence it was not described in the methodology so as not to mislead the reader that we used a standardised questionnaire in our study. The ComPan is not standardised for children.
Our study was concerned with assessing the frequency of consumption of particular food groups, and this is what we focused on. We wanted to verify how much of each food item is consumed by children and adolescents with Down syndrome and we did not examine food diaries. We understand that such a survey would have provided more information, especially when talking about mineral and vitamin deficiencies. However, we took into account that when examining children, examining food diaries was difficult to implement. At the same time, when conducting one of the first such surveys in Poland during the Covid pandemic, we had doubts that the diaries would be filled out correctly without prior face-to-face training with the subjects.
The study group of children and adolescents was divided according to the division taken into account in Polish education. Children up to the age of 2.5 years go to nursery school, then from 3-6.5 years to kindergarten, then from 7 to 10.5 years it is early childhood, then from 11-17.5 years as teenagers and from 18 years of age as adults in Poland. In education, children with Down syndrome are categorised in exactly the same way as healthy children.
We have added information on the statistical test carried out in the statistical analysis of the frequency of consumption in the different age groups.
The study also included people with Down's syndrome who were over the age of 18, although they made up a small percentage of the group, we also wanted to include them in the results because their nutrition did not differ from that of people in their teens. At the same time, such a small number of people in this group is due to the fact that many of them remain in inpatient facilities where they are provided with all-day nutrition, and this fact excluded study participants from the data analysis. Our assumption of the study was that these people would be fed at home.
We have also redrafted the conclusions. We hope that all the changes we have made will be accepted by you.
Thank you again for your time.
Best regards
Agnieszka Białek-Dratwa
Reviewer 2 Report
I would like to thank the authors for submitting their work to “Children”.
Attached please find my comments.
Introduction
Please give more information regarding the cardiovascular risk factors of Down syndrome (dyslipidemia, diabetes type 1/2, physical inactivity, sedentary behaviour, fitness) and how patients could benefit from healthy lifestyle habits.
Please elaborate on nutritional guidelines for children, adolescents and adults. Are there specific recommendations for Down syndrome patients? What nutritional habits are recommended regarding the applied questionnaire (see Figure 1-8)?
Line 42: Change in 1 in 700-1500 live births
Line 48: Language Mistake “tchem”
Line 87: I suggest giving the protein intake as “g per kg bw per day”
Line 88: Please elaborate which “fats” should be limited
Line 128: Please delete – at the end of the sentence
Methodology:
Study group: Was this questionnaire sent to patients in one ore multiple countries? In which country/ies did the patients live?
Exclusion criteria: Was there an age restriction? Which age groups were analyzed (children, adolescents, adults)?
Research tool: Please give a reference of the validated questionnaire. Was the questionnaire translated into another language?
Which questions were opened, which were closed? If closed, please provide answer options.
Line 175-183: Please give more information on the specific questions that were asked. For this, I suggest providing the questionnaire translated into English as a supplementary file.
Line 179 and 181: What is the difference between “coexisting diseases in the examined child” and “co-morbidities of the studied child”. Please elaborate.
Please describe BMI as kg/m2
Results:
The majority of given variables is not introduced in the method section.
Table 2 section associated diseases: What about diseases of other organ systems (e.g. cardiovascular system)? Which form of diabetes is meant?
Table 2 section Breastfeeding: You state that 131 subjects were breastfed. However, when looking at the length of breastfeeding, only 64 subjects were excluded. Please elaborate on that. Were subjects excluded from single analyses?
Do nutritional habits get worse, once patients reach adulthood? I recommend a sub analysis regarding this manner.
Discussion:
The discussion could benefit by comparing the results to previous studies.
Are there results of the used questionnaire for the general population that can be compared with results of the present study?
What nutritional recommendations would you give to parents and patients?
How could we improve nutritional habits (educational programs for parents/patients/care givers etc.)?
Line 330: Source is missing
Study limitations:
The study was conducted during the corona pandemic, potentially this influenced the given results.
Please add that this was a single center study. Potentially your results do not adapt to other countries
References: A large part of references is polish origin. As this article aims for an international audience, I suggest, where applicable, to give English references.
Author Response
Dear Reviewer.
Thank you very much for your time in reading and your indicated suggestions for our study. We have responded to all your suggestions below. In the text we have highlighted any substitutions in yellow to make it easy to follow what we have added.
At the same time, we apologise that you had to wait so long for our response.
We have added information on the cardiovascular risk factors for Down syndrome.
There are no uniform and standardised dietary recommendations for patients with Down syndrome. Nutritional management depends on comorbidities including cardiovascular disease, hypothyroidism and coeliac disease. In each of these disease entities, nutritional management is different. However, it seems very important to develop dietary guidelines for children, adolescents and adults with Down syndrome that cover all the most common comorbidities in Down syndrome.
All comments in the introduction have been corrected.
The survey was conducted among Polish children and adolescents. We have added the phrase 'among Polish'.
The study group of children and adolescents was divided according to the division considered in Polish education. Children up to 2.5 years of age go to nursery school, then from 3-6.5 years of age to kindergarten, then from 7 to 10.5 years of age it is early childhood, then from 11-17.5 years of age as adolescents and from 18 years of age as adults in Poland. In education, children with Down syndrome are categorised in exactly the same way as healthy children.
In the study we did not use the full standardised FFQ, but used some of the questions from it. We relied on the Polish standardised questionnaire KomPAN, however, it is not defined for use among children only for those aged 16 and over, hence it was not described in the methodology so as not to mislead the reader that we used a standardised questionnaire in our study. The ComPan is not standardised for children.
In the study, the open-ended questions were: the exact age of the child, co-morbidities. During the data analysis, we organised the results so that they could be statistically analysed.
We added the survey questionnaire as an appendix.Validation is described in the Research tool subsection.
We decided to remove: "co-morbidities of the study child" as it is duplicative.
Cardiovascular diseases were not entered by parents. Parents had the option to complete the occurrence of diseases.
There was an error in Table 2: Length of breastfeeding 131 children were breastfed, 64 children were not breastfed. 64 children were not breastfed, due to: it was not possible 54 children
no, although it was a possibility 10 children.
And this result was moved incorrectly in the table - we have corrected our mistake. We are very sorry for it AND thank you for bringing it to our attention.
We did not study changes in diet in our group due to growing up. Instead, we observed that there is not much statistically significant change in diet with age.
In the next study, we intend to conduct a study with a control group of healthy children and adolescents. Thank you for the inspiration for the next study in this area, we certainly intend to carry it out.
At the end of the discussion, we also added some recommendations for parents in relation to the nutrition of people with Down's syndrome, but we feel that a separate article should be devoted to this topic, as several of the principles are not sufficient, especially in relation to many associated diseases.
Our study was carried out only in Poland, and it was not a single-city study, but using the CAVI questionnaire, the study was carried out nationwide.
We have internationalised the bibliography, however, some aspects such as dietary standards for Poland we had to leave in the Polish version or references to comparative studies in the Polish population. A few reference items automatically translated into Polish despite being in English. We have corrected this. We apologise for our error.
We have also redrafted the conclusions. We hope that all the changes we have made will be accepted by you.
Thank you again for your time.
Best regards
Agnieszka Białek-Dratwa

Round 2
Reviewer 1 Report
The authors took into account all comments. However, analysis of the survey questionnaire revealed some irregularities that prevented a reliable interpretation of the results. The food frequency measures were different depending on the analyzed food product group.
e.g.
17. How often do you give your child sweets?
· Several times a day
· Once a day
· 4 - 6 times a week
· 2 - 3 times a week · Once a week
· Once every fortnight · Once every three weeks
· Once a month
· Less frequently
· I do not give my child sweets
18. How often does your child eat vegetables?
· Several times a day
· Once a day
· 4 - 6 times a week
· 1 - 3 times a week
· Less often
· Doesn't consumption
How often does your child eat fish?
· Every day
· 5 - 6 times a week
· 3 - 4 times a week
· 1 - 2 times a week
· Once every 2 weeks
· Once every 3 weeks
· Once a month
· Less frequently
· Does not consume
Author Response
Dear Reviewer,
We understand your concerns regarding our questionnaire. The differences in the frequency of consumption of the different food groups are due to the recommendations of the frequency of consumption of the respective products, e.g. the frequency of consumption of vegetables is recommended to be several portions per day, whereas fish, for example, is recommended to be consumed twice a week - hence the differences in the answers.
E.g. if a child eats sweets once a week, or a few times a month - we consider this to be consumption in line with good nutrition. On the other hand, in the case of vegetable consumption, answers e.g. 1-3 times a week mean that vegetable consumption is too low and adding additional answers such as once a month, several times a month is not correct. Hence such differences in the answers.
We hope that we have clarified our answers. and you understand our point of view.
Best regards
Agnieszka Białek-Dratwa
Reviewer 2 Report
several comments were not addressed in the revised manuscript. I still see major limitations in the study design, in the statistical analysis and the conducted conclusions. I therefore suggest to reject this article.
Author Response
Dear Reviewer,
We understand your concerns regarding our publication. However, we do not agree with the statement that our answers did not take into account many aspects. We have responded to each of your suggestions step by step, so please explain what else we can improve so that your publication is accepted. Thank you for your time.
Best regards
Agnieszka Białek-Dratwa
Round 3
Reviewer 1 Report
Thank you for clarifying the method used. Please include this information in the material and methods chapter.
The manuscript requires a slight editorial correction.
Author Response
Dear Reviewer,
We have added a comprehensive description regarding the questionnaire used with a description of the questions used in it. We have highlighted all changes made in green.
We hope that our description satisfies you. And everything will be clear to you. We would like to thank you for all your comments, which have made this article more comprehensible to the reader.
We send our best regards and wish you a good day.
Authors
